# Green Line Hospital-Territory Study: A Single-Blind Randomized Clinical Trial for Evaluation of Technological Challenges of Continuous Wireless Monitoring in Internal Medicine, Preliminary Results

**DOI:** 10.3390/ijerph181910328

**Published:** 2021-09-30

**Authors:** Filomena Pietrantonio, Antonio Vinci, Francesco Rosiello, Elena Alessi, Matteo Pascucci, Marianna Rainone, Michela Delli Castelli, Angela Ciamei, Fabrizio Montagnese, Roberto D’Amico, Antonella Valerio, Dario Manfellotto

**Affiliations:** 1Internal Medicine Unit, Ospedale dei Castelli, ASL Roma 6, Va Nettunense, 00043 Rome, Italy; filomena.pietrantonio@aslroma6.it (F.P.); elena.alessi@aslroma6.it (E.A.); matteo.pascucci@aslroma6.it (M.P.); marianna.rainone@aslroma6.it (M.R.); michela.dellicastelli@aslroma6.it (M.D.C.); angela.ciamei@aslroma6.it (A.C.); fabrizio.montagnese@aslroma6.it (F.M.); 2School of Hygiene and Preventive Medicine, University of Rome “Tor Vergata”, 00100 Rome, Italy; antonio.vinci.at@hotmail.it; 3Department of Infectious Disease and Public Health, Sapienza University of Rome, P. le A Moro, 5, 00185 Roma, Italy; 4Centro Oncologico Modenese, Department of Medical and Surgical Sciences, Mother and Child and Adult Health, Modena and Reggio Emilia University, 3° Piano, Azienda Ospedaliero-Universitaria Policlinico di Modena, Via del Pozzo 71, 41124 Modena, Italy; roberto.damico@unimore.it; 5Fadoi Foundation, Piazza Cadorna 15, 20123 Milano, Italy; antonella.valerio@fadoi.org; 6Internal Medicine Unit, Fatebenefratelli Isola Tiberina, Via di Ponte Quattro Capi 39, 00186 Roma, Italy; dario.manfellotto@afar.it

**Keywords:** wireless monitoring system, telemedicine, integrating hospital and community, acute medicine, poly-morbidity, internal medicine core competencies

## Abstract

Background: Wireless vital parameter continuous monitoring (WVPCM) after discharge is compared to regular monitoring to provide data on the clinical-economic impact of complex patients (CPs) discharged from Internal Medicine Units of Ospedale dei Castelli, Lazio. Primary outcome: Major complications (MC) reduction. Secondary outcomes: Patients who reached discharge criteria within the 7th day from admission; difference in MC incidence at the conclusion of the standard telemonitoring/clinical monitoring phase, 5 and 30 days after discharge; and conditions predisposing to MC occurrence. Methods: Open label randomized controlled trial with wearable wireless system that creates alerts on portable devices. Continuous glycemic monitoring is performed for patients with diabetes mellitus. Results: There were 110 patients enrolled (mean age: 76.2 years). Comorbidity: Cumulative Illness Rating Scale CIRS-CI (comorbidities index): 3.93, CIRS SI (severity index): 1.93. About 19% scored a BRASS (Blaylock Risk Assessment Screening Score) ≥20 indicating need for discharge planning requiring step-down care. Globally, 48% of patients in the control group had major complications (27 out of 56 patients), in contrast to 22% in the intervention group (12 out of 54 patients). Conclusions: Since WVPCM detects early complications during the post-discharge CPs monitoring, it increases safety and reduces inappropriate access to the Emergency Room, preventing avoidable re-hospitalizations.

## 1. Introduction

In recent years, more and more patients admitted to internal medicine (IM) wards have been reported as being in large part affected by a wide range of acute, subacute, or chronic diseases, all in different grades of severity. Often the patients are elderly, frail, and with multiple co-morbidities, requiring intensive care treatment [1]. Hospitalization of such patients in large wards without prior stratification of severity, complexity, level of dependence, co-morbidities and, moreover, without a proper risk assessment for rapid clinical deterioration, has been demonstrated to be a major cause of suboptimal treatment, leading to an increase in hospital length of stay and rising care costs [2]. Recent literature data show that 27% of patients admitted to IM wards are critical and therefore in need of continuous monitoring and advanced medical technology. Furthermore, 25% of hospitalized patients had social difficulties requiring the activation of integrated hospital–community pathways [3]. The definition of the roles of hospital and community, in response to patient needs, is the fundamental component of the socio-health organizational models that is being implemented in various Italian regions. In this context it is necessary to define the most appropriate care settings, such as the care for standard patient management, the technological requirements, and the crucial pathways. The epidemiological transition has caused a constant increase in aged patients and comorbidities of patients seeking for urgent access to hospital care and were in need of hospitalization in the medical area. To optimize the allocation and the organization of the resources, new management models should be implemented. These models should use mobile devices to monitor patients with the goal of reducing clinical risk, ensuring patient safety, and minimizing costs. wireless vital parameter continuous monitoring (WVPCM) can allow us to promptly identify and highlight the worsening of clinical conditions in complex patients (CPs) who are not subject to intensive care monitoring, such as those admitted to the IM. The interventions of health personnel can thus be directed towards patients who need immediate assistance, allowing a correct and targeted intervention in a very short time. For patients with diabetes mellitus, continuous glucose monitoring (CGM) may consist of a reduction in hemoglobin A1c (HbA1c), time spent in hypoglycemia., and acute and long-term complications [4,5]. The evidence in the literature of major complications and/or adverse events that may occur in patients at home does not appear homogeneous regarding the duration of the observation, the type of intervention performed to control complications, and the temporal relationships with respect to hospitalizations [6,7]. Very different results are reported between studies, ranging from 3.5% to 15.1%. [8] Factors that are predisposed to adverse events are old age, co-morbidities, sex, depression, cognitive impairment, functional limitations, reduced compliance, and a lack of a caregiver [9,10]; furthermore, there are some telemedicine randomized trials available in the current literature, but they investigate only a single outcome [11,12]. When including patients with comorbidities, they provided moderate evidence of improvements in measures of disease control, but little evidence and no demonstrated benefits on health status. Further research is needed with clear descriptions of conditions, interventions, and outcomes based on patients’ and healthcare providers’ preferences [13]. The continuous monitoring of vital signs may allow the detection of early signs of clinical worsening even of patients in a non-intensive care setting (including at home or within subacute facilities). It also allows the medical team to immediately take note of any variation in the patient’s clinical conditions and to deliver the most appropriate therapy.

## 2. Study Objective

This study protocol was registered on ClinicalTrials.gov (Identifier: NCT03764709) and approved by the Ethics Committee “Roma 2” on 12 September 2018, as protocol n° 112.18. The first patient was enrolled on 23 September 2019.

This work investigates the efficacy of wireless vital parameter continuous monitoring (WVPCM) in reducing incidence of major complications and in improving the outcome and the quality of care in stable patients discharged from the Internal Medicine Unit. It also assesses any variation in hospitalization costs, for instance by decreasing any inappropriate access to the emergency room, and (as a proxy parameter) by preventing avoidable re-hospitalizations, when compared to the current standard of care (conventional monitoring). Its aim is to investigate the effectiveness of the WVPCM versus the current standard of care (conventional monitoring) in the first 5 days after discharge from the Internal Medicine Units for stable, complex patients regardless of the reason for hospitalization.

## 3. Materials and Methods

### 3.1. Study Design

The FADOI Foundation (Italian Scientific Society of Internal Medicine) (co-founder), with the collaborative support of ASL (Local Health Authority) Roma 6 (founder), promoted a multicenter open label randomized controlled trial on patients admitted to the IM and subsequently dismissed, either to a low-intensity care unit or at home in a protected discharge. This study has been reported using the CONSORT tool [14] (Appendix A).

The Greenline H-T study is one of the first multicenter open label randomized controlled trials [15] aimed at evaluating the management of complex patients hospitalized in the Internal Medicine Unit in the early phase of discharge by using the WVPCM (WMON arm) and comparing it to conventional monitoring (STD) (Figure 1).

This also evaluates the effectiveness of the remote monitoring for 5 days compared to the traditional clinical monitoring after discharge. Two patient settings denominated by Group A and Group B were observed. Group A included patients admitted to the acute ward who were considered transferable to a low-intensity ward within 7 days from admission. Group B included patients admitted to the acute ward who were considered dischargeable to home within 7 days from admission.

### 3.2. Study Arms

(1)Experimental group: all patients with inclusion criteria randomized by trial software.(2)Control group: all patients with inclusion criteria randomized by trial software.
The study will involve the enrollment of 300 patients. Among these, 150 patients will undergo continuous monitoring and 150 will undergo controls.

### 3.3. Identification of the Monitoring Tool and Procedure

The WIN@Hospital system (WINMEDICAL, Navacchio, (PI), Italy) is a portable wireless system (Medical Class IIA) that allows continuous, real-time vital signs monitoring, automatic calculation of the NEWS (National Early Warning Score) [16] score, and the creation of a personalized alert system for every single patient through a portable device (tablet or phone), without the necessity of a constant presence of nursing staff. Monitored vital signs are cardiac rate (expressed in beats per minute (BPM), beats/minute), respiratory rate (in acts/minute), blood pressure (in mmHg, millimeters of mercury), peripheral saturation (percentage of oxygen-binding hemoglobin), temperature (in Celsius grade), and position of the patient (orthostatism/clinostats). The WIN@Hospital system is assembled by one module for every parameter, core unit, battery, and wireless connection unit. Overall, the dimensions are 2.5 × 2 × 10 cm (thickness × height × width), and the weight is 0.2 kg. By web app, it is possible to customize alerts for every parameter and for every patient before and during monitoring. A preview pilot study was conducted in 2019 in the internal medicine ward of Ospedale dei Castelli (ASL Roma 6, Lazio) and the territorial department of ASL Roma 6. A web app was provided to randomize the patients (clinician does not know randomization criteria) based on their age, sex, weight, and pathologies. Technical specifications, as provided by the manufacturer, are provided in Appendix A.

A flowchart depicting a detailed explanation of the procedure used on home discharge is shown in Figure 2.

### 3.4. Safety Assessment

The WIN@Medical monitoring systems display data in real time, but they do not require nursing staff to constantly monitor the data. As a class IIA device, it is not designated as an intensive monitoring tool nor as a life-saving tool (class II B), but as a simple support tool aimed at tracking clinical parameters. In order to guarantee privacy safety, the architecture of digital infrastructure is closed (Figure 3).

### 3.5. Time Frame

The first patient was enrolled on 23 September 2019. The main study is still ongoing and is planned to remain active for 24 months from the first enrollment.

### 3.6. Primary Outcomes

Reduction in major complications (Table 1).

### 3.7. Secondary Outcomes

A percentage of patients admitted to IM met the discharge criteria (home or transfer to a subacute ward) after 7 days from admission; differences in MC incidence at the conclusion of the telemonitoring/ standard clinical monitoring 5 and 30 days after discharge; conditions predisposing patients to the MC occurrence; differences in re-hospitalization rates.

### 3.8. Statistical Methodology

Observing the characteristics of patients in internal medicine, the trend of complexity and comorbidities increased [17], and it is likely it will result in an increase in both intra-hospital and home complications.

For these reasons it is difficult to hypothesize a percentage incidence value for major complications and/or adverse events as a base to calculate the sample size to be observed. It appears essential to carry out a pilot study to evaluate the real incidence of complications at the time of hospital discharge. In addition to that, it will be important to evaluate the feasibility of using monitoring systems that can minimize adverse events by an early intercepting of clinical conditions deterioration.

The scientific literature appears to be particularly heterogeneous with regards to the occurrence of complications at home, the type of complications themselves, the acute, post-acute, or chronic condition of the patient, and finally, the duration of the observation. In addition to this, no information is available on the potential effects of continuous monitoring of vital signs, particularly for a patient discharged from internal medicine wards.

In this perspective, it is problematic to formulate hypotheses on the incidence of complications in the group of patients undergoing continuous monitoring and in the control group. Therefore, it seems reasonable to make a project that does not include a formal calculation of the sample size but, due to its prospective, randomized, and controlled nature, will allow a reliable definition of the clinical scenario.

Based on a feasibility criterion, an overall enrollment of 300 patients (150 patients undergoing continuous monitoring and 150 undergoing standard clinical monitoring) was estimated over a period of approximately 2 years. Randomization will be carried out with stratification between subgroups “discharged at home” and “transferred to the low intensity ward”, with a ratio of 2:1 between the two subgroups (2 patients Group B, 1 patient Group A).

### 3.9. Statistical Analysis

This study was designed as a multicenter open label randomized controlled trial [18]. Due to difficulties arising from COVID-19 syndemics, only Ospedale dei Castelli (NOC) currently was able to enroll patients for this study.

Data were collected by medical personnel in Ospedale dei Castelli in a cloud platform done by the University of Modena and Reggio Emilia. Statistical personnel of the University of Modena and Reggio Emilia carried out the statistical analysis.

Univariate comparisons of data were performed using the Student t test for continuous variables and the Yates chi-square for categorical variables. Adjusted means were calculated as well as mean differences, with corresponding confidence intervals, between the two arms. The covariates included in the model were age, gender, CIRS, BRASS, Barthel score, Exton–Smith Scale, Charlson Index, and patient origin. All statistical analyses were conducted using SAS v.9.2 (SAS Institute Inc., Cary, NC, USA).

### 3.10. The Rationale for Monitoring

Hospitalized patients should be adequately monitored with attention for abnormal physiological signals that may precede significant deterioration. In recent clinical trials it has been shown that death and Intensive Care Unit (ICU) transfer can be avoided in critically ill patients if the patient’s deteriorating clinical conditions are noted early. According to a recent Italian retrospective multicenter study, the overall average incidence of adverse events during hospitalization is 5.2% (lower value than the data of international studies, which is 9.2%), and adverse events are prevalent in medical areas (37.5%). The study found a total of nationally preventable events equal to 56.7%. The prevailing consequence of adverse events appears to be the prolongation of hospitalization, followed by the presence of a disability at the time of discharge, while the patient’s death has a median occurrence of 9.5% [19,20]. In the United States, it is estimated that the costs of compensation related to adverse events amount annually to about 17 to 29 billion dollars. In Italy, the costs associated with claims in 2002 were estimated to amount to approximately €2.5 billion/year. The cost of claims for healthcare facilities is approximately €107 per hospitalization (estimated March 2015, for the decade 2004–2013) [21].

### 3.11. Wireless Monitoring System

Preliminary studies on patients wireless monitoring in ordinary hospitalization [22,23] have shown that, through the monitoring and the early detection of clinical instability symptoms, the average hospital stay is reduced, allowing significant reductions in terms of hospitalization costs. The main expected results in the use of a monitoring system are the following [24]:possible reduction in length of hospital stay and safe early discharge;reduction in ICU transfers;early transfer of patients from areas with higher-intensity care to areas with lower-intensity care and an estimated savings of €290 per day for each ordinary patient who is transferred from an intensive ward to a lower-intensity ward;a nurse reduction in time dedicated to the detection of vital parameters and the potential employ of the nurse’s skill in more valuable activities.

### 3.12. Inclusion/Exclusion Criteria

Patients are included or excluded from participating in the study according to a set of pre-defined criteria, reported in Table 2.

### 3.13. Randomization Criteria

Sequence generation:

Permuted-blocks (block sizes of 4 and 6) randomization sequences were generated by a statistician by using the Stata^®^ 12.0 statistical software, stratified by age group and type of discharge.

After the clinician had evaluated inclusion criteria and obtained the patient’s consent, they connected to a dedicated secure web-based elettronic data collection sheet (eCRF) system for patient registration, fulfilling the screening form and randomization procedure. At the end of the randomization procedure, the system returned the allocation arm based on the randomization sequences. Patients were showed as “NOCxx”, where “xx” is a progressive sequential number.

Only the statistician and the web-based system administrator were aware of the randomization sequences.

Only the clinicians were aware of patient data (name, surname, date of birth, etc.).

## 4. Results

Since September 2019, 110 patients were enrolled (56 M/54 F; 56 in experimental arm and 54 in control one; 100% discharged home); average age was 76.2 (range 43–92, 75.9 in STD and 76.7 in WMON arm, with 45 patients (42.5%) aged 80 years or older). Flow of enrolled patients is shown in Figure 4.

CIRS-CI (Comorbidity: complexity index) average value was 3.93 (range 0–13), CIRS SI (severity index) 1.93 (range 1.1–3.1). Twenty patients (18,9%) scored BRASS (Blaylock Risk Assessment Screening Score) ≥ 20, average 12.8 (range 0–28); Barthel mean value 65.2 (range 0–100); Exton–Smith scale 16.5 (range 7–20); average Charlson Index 4,0 (range 0–93.3), indicating need for step-down care. Twenty-four patients (21.81%) had major complications (MC) at 5 days and 25 (22.72%) at 30 days of follow-up.

The main complications highlighted by continuous home monitoring were cardiac arrhythmias, glycemic decompensation, and drug interactions.

Current findings of the Greenline H-T study are reported in Table 3.

Five-days follow up (Figure 5) shows 24 patients with MC: 1 stroke (STD arm), 4 transient ischemic attacks (3 STD, 1 WMON), 1 angina pectoris (STD), 2 downfalls (1 STD, 1 WMON), 4 hospital infections (2 STD, 2 WMON), 2 delirium (STD), 8 adverse drug reactions (4 STD, 4 WMON), 2 bedsores (WMON), 1 urinary catheter (STD), and 7 re-hospitalizations (5 STD–2 WMON).

Thirty-days follow up (Figure 6) shows 25 patients with MC: 4 deaths (3 STD, 1 WMON), 3 transient ischemic attacks (WMON), 2 heart attacks (STD), 3 nosocomial infections (STD), 2 downfalls (1 STD, 1 WMON), 1 venous thrombosis (WMON), 1 angina pectoris (STD), 2 delirium (1 STD, 1 WMON), 14 adverse drugs reactions (11 STD, 3 WMON), 1 artificial nutrition (STD), 3 bedsores (2 STD, 1 WMON), 2 urinary catheters (1 STD, 1 WMON), 1 arrhythmia (STD), and 14 re-hospitalizations (10 STD, 4 WMON). One patient had some episodes of glycemic decompensation.

## 5. Discussion

Recent studies have shown a reduction of up to 60% (345 min per day) [25] of nurse care with telemedicine systems. Since there are no studies in current literature comparing the use of wireless monitoring systems of vital signs in complex patients to a more traditional monitoring system, the intent of this work is to help highlight the potential benefits of a continuous monitoring strategy in terms of reduction in major complications, improvement of outcomes, and reduction in costs. So far, the initial results (calculated on 110 patients, representing almost 37% of the estimated total sample size), in terms of reduction in major complications, seem to encourage the use of continuous monitoring over traditional monitoring. In fact, reductions in the frequency of falls and in adverse reactions in the study group were found, compared to the control group. In addition, most complications or situations posing danger for a patient’s life (TIA, heart attack, glycemic decompensation) have been observed only in the control group and have not happened so far in the intervention group. Globally, 48% of patients in the control group have had some major complications (27 out of 56 patients), in contrast to 22% in the intervention group (12 out of 54 patients).

Preliminary results concerning the primary endpoint show a reduction in major complications incidence, since fewer of them were recorded in the experimental group than in the control group.

Regarding the secondary endpoint, we have highlighted that the experimental arm has half the hospital re-admissions rate compared to the control group.

### Limitations of the Study

The aim of this article is to represent the study design and the preliminary results of the Greenline H-T study. Since the number of patients recruited represents one-third of the expected sample, we have only reported the trends highlighted by the analysis of the preliminary results. Due to the COVID-19 pandemic, there has been a slowdown in recruitment which was performed only at the NOC Hospital despite the whole structure having been completely converted to a COVID-19 hospital for over 6 months [26]. Randomization is useful to avoid bias and presence of the same number of patient-independent complications to confirm the internal validity of the study, despite the patient being unblinded regarding his collocation in the study arms.

## 6. Conclusions

The WVPCM allows the medical team to detect early any variation in clinical conditions of complex patients and to promptly perform diagnostic and/or therapeutic interventions, increasing patient safety and reducing care costs. Extending the wireless monitoring system to low-intensity facilities and even domestic settings may both guarantee expert assistance after hospital discharge and reduce overcrowding in emergency departments and hospital wards. Doing the right thing to the right patient at the right time represents the core mantra that supports a high-quality diagnostic/therapeutic pathway that, along with appropriate and efficient clinical reasoning, is the best way to reduce costs and hospital overcrowding while delivering state-of-the-art medical care. Integrating hospital and community is a new challenge for telemedicine that allows for improving patient management both during hospitalization and after discharge. Following the COVID-19 pandemic, the use of telemedicine appears increasingly suitable for remote monitoring of fragile patients. The hospital-community integration model proposed by the Greenline H-T study appears to be very promising to increase the safety of discharge, improve the control of patients at home, and reduce the need for emergency services and the incidence of inappropriate hospitalizations.

## Figures and Tables

**Figure 1 ijerph-18-10328-f001:**
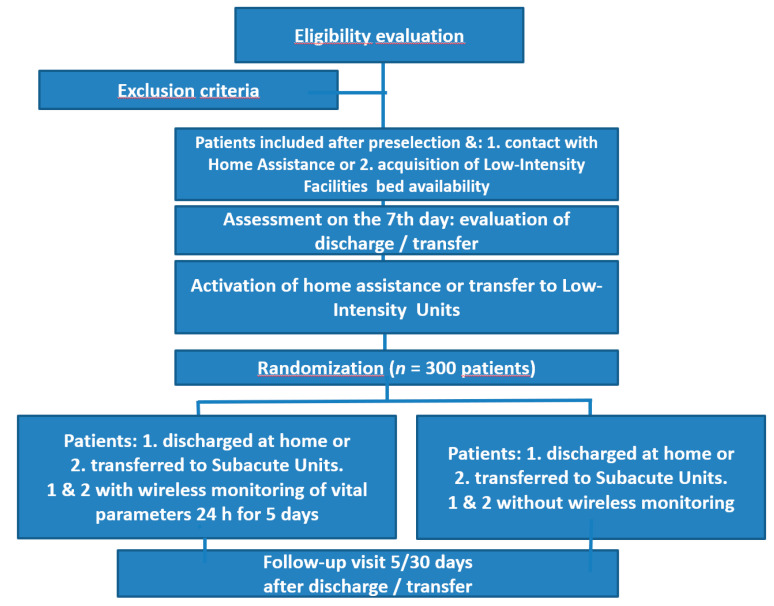
Study Design (modified from Moher D, Hopewell S, Shulz KF, et al.) [12].

**Figure 2 ijerph-18-10328-f002:**
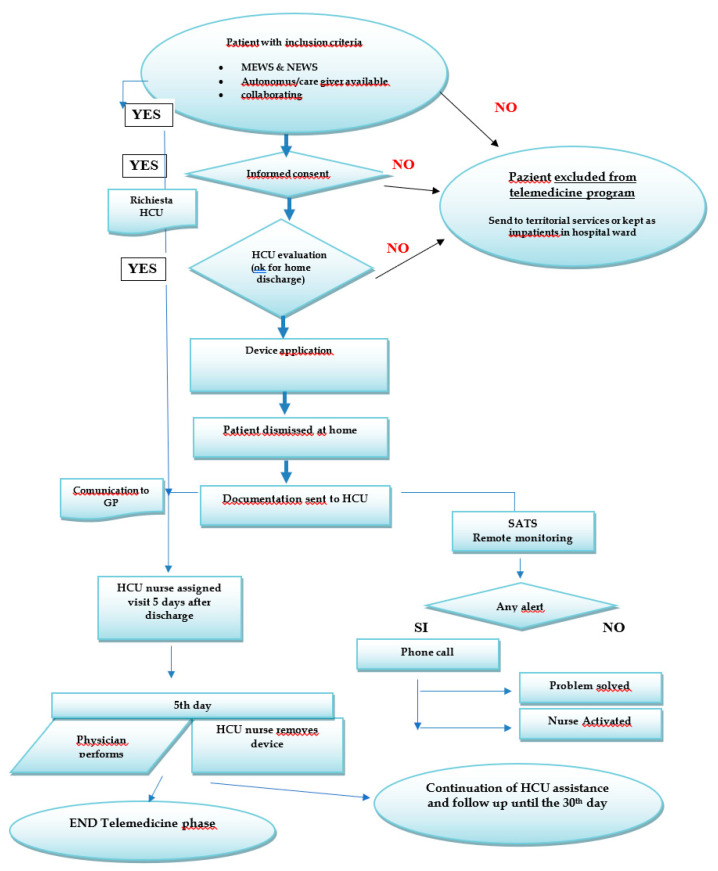
Telemedicine procedure flowchart. HCU: Health Care Unit, GP: General Practitioner, SATS: Telemedicine Operation Centre, MEWS: Modified Early Warming Score.

**Figure 3 ijerph-18-10328-f003:**
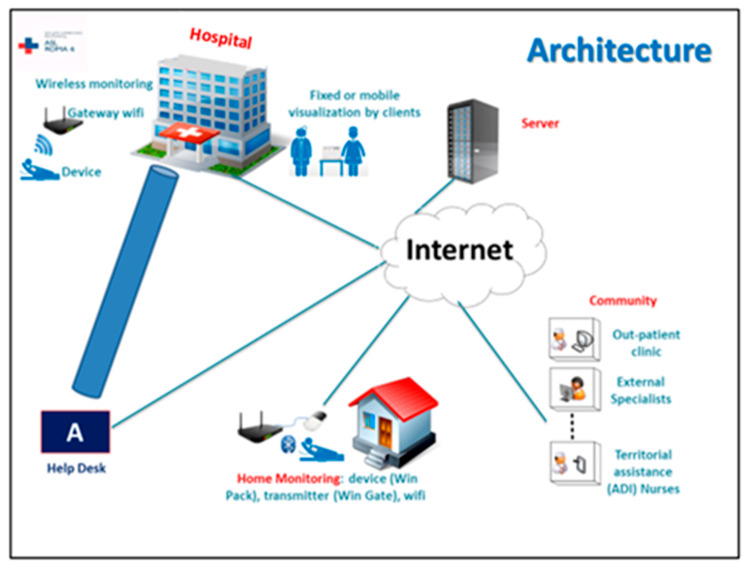
Architecture of digital infrastructure.

**Figure 4 ijerph-18-10328-f004:**
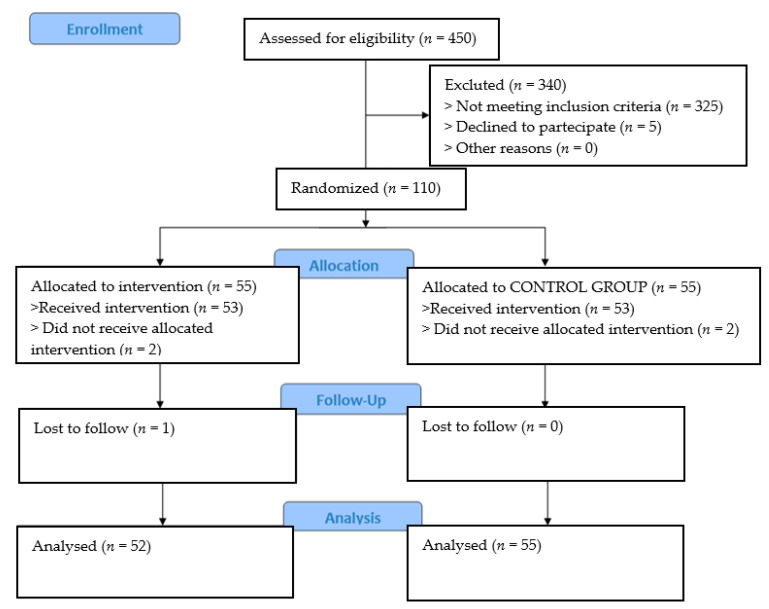
Patients’ enrollment flow.

**Figure 5 ijerph-18-10328-f005:**
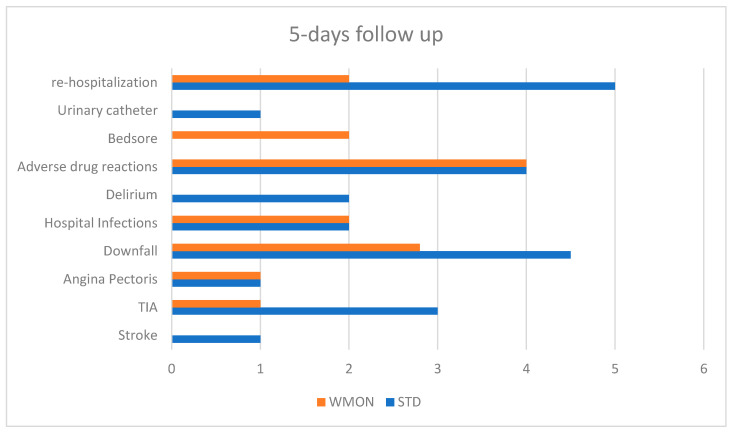
5-days follow up. STD: control arm; WMON: wireless monitoring arm. TIA: transient ischemic attack.

**Figure 6 ijerph-18-10328-f006:**
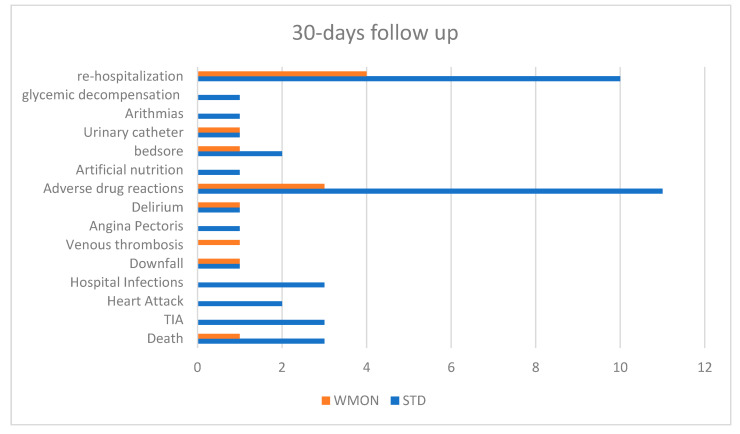
30-days follow up. STD: control arm; WMON: wireless monitoring arm. TIA: transient ischemic attack.

**Table 1 ijerph-18-10328-t001:** Major complications, adopted (modified) by Agency for Healthcare research and Quality (AHrQ) [15].

Major Complications	
1.	Unexpected death
2.	Cardiac arrest
3.	Healthcare-related infections (urinary tract infections, pneumonia, sepsis, etc.)
4.	Cardiac arrhythmias, acute coronary syndrome
5.	Acute renal failure
6.	Acute respiratory failure
7.	Onset and evolution of neurological deficits not present at admission
8.	Deep vein thrombosis and pulmonary thromboembolism
9.	Gastrointestinal hemorrhage
10.	Pressure ulcers
11.	Injury/fall
12.	Allergic reactions
13.	Unexpected transfer from a general care ward to a more addictive/intensive care ward
14.	Unplanned new hospitalization within 21 days of discharge
15.	Episodes of glucose decompensation (hypo/hyperglycemia) in patients with diabetes mellitus.

**Table 2 ijerph-18-10328-t002:** Inclusion and exclusion criteria.

Inclusion Criteria	Patient > 18 years old, AND Patient hospitalized in the Internal Medicine Unit in stable clinical condition (normal MEWS and NEWS scores), AND Patient presumed dischargeable on the seventh day of admission regardless of the hospitalization reason with BRASS > 11 and almost 2 active pathologies, AND Patient agreed to participate in the study before discharge, AND Patient (or any co-living caregiver) was able to use the device.
Exclusion Criteria	Patient with COVID-19 diagnosis, OR Patient living in low-intensity facilities Patient with unstable conditions (terminal cancer, severe cognitive impairment or not dischargeable on the seventh day), OR Patient did not agree/was unable to express valid consent to partecipate in the study, OR Patient (or any co-living caregiver) was not able to use the device.

**Table 3 ijerph-18-10328-t003:** Characteristics of enrolled patients.

Items	Preliminary Results	Observations
Patients enrolled	110 (56 M/54 F)	Enrollments in progress
Mean Age	76.2 years (range 43–92)— STD 75.9, WMON 76.7	45 patients (42.5%) > 80 years
Comorbidity	CIRS CI 3.93 CIRS SI 1.93	High patient complexity
BRASS ≥ 20	20 patients (18.9%)	Indicating need for step-down care
Barthel Score	65.2 (range 0–100)	Mean value
Exton–Smith scale	16.5 (range 7–20)	Mean value
Charlson Index	3.8	Mean value
Major Complications at 5 days of follow up	24 (14 STD, 10 WMON)	25% STD, 19% WMON Favors intervention
Major Complications at 30 days of follow up	39 (27 STD, 12 WMON)	48% STD, 22% WMON
Re-hospitalization in 5 days	7 (5 STD, 2 WMON)	Favors intervention
Re-hospitalization in 30 days	14 (10 STD, 4 WMON)	WMON arm has half hospitalization rate

M/F: Male/Female. BRASS: Blaylock Risk Assessment Screening Score, CIRS: Cumulative Illness Rating Scale, STD: control arm, WMON: wireless monitoring arm.

## Data Availability

Data available on request due to restrictions (privacy).

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
