# Peer review of "Green Line Hospital-Territory Study: A Single-Blind Randomized Clinical Trial for Evaluation of Technological Challenges of Continuous Wireless Monitoring in Internal Medicine, Preliminary Results"

_ijerph, 2021, doi:10.3390/ijerph181910328_

Round 1

Reviewer 1 Report

The reviewed work presents the methodology and some preliminary results on the efficacy of a Wireless Vital Parameter Continuous Monitoring in reducing incidence of major complications and in improving the outcome and the quality of care in stable patients discharged from Internal Medicine Unit. The wireless system collects the cardiac rate, respiratory rate, blood pressure, peripheral saturation , temperature  and patient position. 

This is a nice work, and have been conducted correctly. It is a relevant theme, with applicability in practically all medical services, but it is in an initial phase. 
The work presents few results and does not present evidences of their significance. Increasing data volume seems to be mandatory, in order to demonstrate such significance.
Authors claim that it is difficult to define a percentage incidence value for major complications and/or adverse events as a base to calculate the sample size to be observed, but it is important to quantify results with statistical support. 

Some detailed comments:

Line 134 – “Overall, the dimensions are 2,5*12*10 (thick- 134 ness*height*width) and the weight is 0,2 kg.” Please, provide units (cm?)
Line 135 – Language review: it is possible to customize alert for every parameter and for every patient before and during monitoring

Author Response

This is a nice work, and have been conducted correctly. It is a relevant theme, with applicability in practically all medical services, but it is in an initial phase. The work presents few results and does not present evidences of their significance. Increasing data volume seems to be mandatory, in order to demonstrate such significance. Authors claim that it is difficult to define a percentage incidence value for major complications and/or adverse events as a base to calculate the sample size to be observed, but it is important to quantify results with statistical support.

Some detailed comments:

Line 134 – “Overall, the dimensions are 2,5*12*10 (thick- 134 ness*height*width) and the weight is 0,2 kg.” Please, provide units (cm?) Line 135 – Language review: it is possible to customize alert for every parameter and for every patient before and during monitoring

=====================================

Line 134 and 135 have been corrected.

The work presents few results and does not present evidences of their significance. Increasing data volume seems to be mandatory, in order to demonstrate such significance.”

Actually, this are the preliminary result on a 37% of total sample of a pilot study. This has been explicated in the text. The evidence in the literature of major complications and/or adverse events that may occur in patients at home does not appear homogeneous regarding the duration of the observation, the type of intervention performed to control complications and the temporal relationship with respect to hospitalization (6), (7). Very different results are reported between studies, ranging from 3.5% to 15.1%. (8)

Blais R, Sears NA, Doran D, Baker GR, Macdonald M, Mitchell L, et al. Assessing adverse events among home care clients in three Canadian provinces using chart review. BMJ Qual Saf. 2013 Dec;22(12):989–97.

Sears N, Baker GR, Barnsley J, Shortt S. The incidence of adverse events among home care patients. Int J Qual Health Care. 2013 Feb 1;25(1):16–28.

Masotti P, McColl MA, Green M. Adverse events experienced by homecare patients: a scoping review of the literature. Int J Qual Health Care. 2010 Apr 1;22(2):115–25.

Reviewer 2 Report

Hhello
The manuscript deals with an important topic nowadays. I suggest some fundamental adjustments so that you can move forward for future publication.

Insert in the title that it is a randomized clinical trial;
In the summary: Please enter the place of collection as well as the date the data were collected.
Insert the ECR registration number after the summary;
In the method: Please remove the objective found in this section and insert it after the introduction;
He favors mentioning what he did with as "losses";
Mention the exclusion criteria, they are important in randomized clinical trials;
In the results: Please insert the flow of participants (*important*)
Insert as of recruitment dates and follow-up periods;
Show the number of losses you incurred during follow-up and your reasons;
In the images, you are sure to bring the result, I suggest that you do not know that the study has not yet been completed. Be more cautious in generalizing, as well as in the external validity and applicability of the findings of this clinical study;
Please indicate who writes the study according to the Consort tool. 

Author Response

The manuscript deals with an important topic nowadays. I suggest some fundamental adjustments so that you can move forward for future publication.

Insert in the title that it is a randomized clinical trial; In the summary: Please enter the place of collection as well as the date the data were collected. Insert the ECR registration number after the summary; In the method: Please remove the objective found in this section and insert it after the introduction; He favors mentioning what he did with as "losses"; Mention the exclusion criteria, they are important in randomized clinical trials; In the results: Please insert the flow of participants (*important*) Insert as of recruitment dates and follow-up periods; Show the number of losses you incurred during follow-up and your reasons; In the images, you are sure to bring the result, I suggest that you do not know that the study has not yet been completed. Be more cautious in generalizing, as well as in the external validity and applicability of the findings of this clinical study; Please indicate who writes the study according to the Consort tool.

=================================

Title has been corrected as suggested; Data collection place has been inserted in the summary; ECR registration number has been inserted after the summary; Objective section has been added; exclusion and inclusion criteria table has been added; flow of participant figure and numbers have been added; dates have been declared; CONSORT tool has been mentioned and used in study reporting, and its checklist added to supplementary materials.

Reviewer 3 Report

Dear authors 

The article has an interesting topic. There is no detail in most key parts of the article. All sections have been presented with a shallow structure. Many authors are participating in the study, and we expect a well-design study with perfect details. There is no managerial implication. Study design needs way more explanations. The lack of literature review part is undeniable. 

Author Response

The article has an interesting topic. There is no detail in most key parts of the article. All sections have been presented with a shallow structure. Many authors are participating in the study, and we expect a well-design study with perfect details. There is no managerial implication. Study design needs way more explanations. The lack of literature review part is undeniable. 

=================================

Actually, this are the preliminary result on a 37% of total sample of a pilot study. This has been explicated in the text. The evidence in the literature of major complications and/or adverse events that may occur in patients at home does not appear homogeneous regarding the duration of the observation, the type of intervention performed to control complications and the temporal relationship with respect to hospitalization (6), (7). Very different results are reported between studies, ranging from 3.5% to 15.1%. (8)

Several details have been added in results reporting. Procedures and protocols have been explicated, either in text or in flowcharts within figures. Study objectives and design sections have been further expanded. Bibliography has been expanded as well.

Title has been corrected as suggested; Data collection place has been inserted in the summary; ECR registration number has been inserted after the summary; Objective section has been added; exclusion and inclusion criteria table has been added; flow of participant figure and numbers have been added; dates have been declared; CONSORT tool has been mentioned and used in study reporting, and its checklist added to supplementary materials.

Reviewer 4 Report

The discharge of patients to the home, especially those who may be unstable, is a health and economic problem. Extending admission for a few more days to be more certain that the patient is not going to become more complicated, forces the use of hospital beds that could be used for other purposes. The possibility of home monitoring of these patients would be an alternative.
The article submitted for review is interesting and addresses this problem. I see as a limitation of the study the small number of patients and it would be interesting if the study were multicenter.
On the other hand, the article explains very little about the patient monitoring mechanism and its technical characteristics. I believe that the authors should specify this point more in the article before it is accepted for publication.

Author Response

The discharge of patients to the home, especially those who may be unstable, is a health and economic problem. Extending admission for a few more days to be more certain that the patient is not going to become more complicated, forces the use of hospital beds that could be used for other purposes. The possibility of home monitoring of these patients would be an alternative. The article submitted for review is interesting and addresses this problem. I see as a limitation of the study the small number of patients and it would be interesting if the study were multicenter. On the other hand, the article explains very little about the patient monitoring mechanism and its technical characteristics. I believe that the authors should specify this point more in the article before it is accepted for publication.

=================================

The study design and protocol is actually a multi-centre one. Due to COVID pandemic, only one involved center managed to actually perform enrollment, but it is potentially still being carried by other centers. This has been explicated in text.

Technical documentation has been uploaded as supplementary material. Telemedicine procedure has been explicated in flowchart figures.

Round 2

Reviewer 1 Report

The authors addressed most, but not all, reviewers' comments. Although this is a preliminary article, I still think the authors should provide evidence of the significance of the data. Some analysis of data significance is always desirable when the study has a low volume of data. Despite this, I believe the work has improved a lot and can be published in the journal.

Reviewer 2 Report

dear, hello! thank you again for the opportunity to review the manuscript.
The authors made some adjustments, but not all. I identify that this is not something that discredits the manuscript, as the main points have been clarified (reasons for the losses and insertion of a flowchart).
I'm available,

This manuscript is a resubmission of an earlier submission. The following is a list of the peer review reports and author responses from that submission.